# Ovarian Cancer Dissemination—A Cell Biologist’s Perspective

**DOI:** 10.3390/cancers11121957

**Published:** 2019-12-06

**Authors:** Sadaf Farsinejad, Thomas Cattabiani, Taru Muranen, Marcin Iwanicki

**Affiliations:** 1Department of Chemistry and Chemical Biology, Stevens Institute of Technology, Hoboken, NJ 07030, USA; sfarsine@stevens.edu (S.F.); tcattabi@stevens.edu (T.C.); 2Department of Medicine, Harvard Medical School, Boston, MA 02115, USA; tmuranen@bidmc.harvard.edu

**Keywords:** ovarian cancer, fallopian tube secretory epithelial cells, peritoneal dissemination, myosin, integrins

## Abstract

Epithelial ovarian cancer (EOC) comprises multiple disease states representing a variety of distinct tumors that, irrespective of tissue of origin, genetic aberrations and pathological features, share common patterns of dissemination to the peritoneal cavity. EOC peritoneal dissemination is a stepwise process that includes the formation of malignant outgrowths that detach and establish widespread peritoneal metastases through adhesion to serosal membranes. The cell biology associated with outgrowth formation, detachment, and de novo adhesion is at the nexus of diverse genetic backgrounds that characterize the disease. Development of treatment for metastatic disease will require detailed characterization of cellular processes involved in each step of EOC peritoneal dissemination. This article offers a review of the literature that relates to the current stage of knowledge about distinct steps of EOC peritoneal dissemination, with emphasis on the cell biology aspects of the process.

## 1. Introduction

### 1.1. EOC Classification

EOC is the most common form of ovarian cancer, and it can be divided into two types based on histology. Type I tumors include endometroid, mucinous, clear-cell, and low-grade serous carcinomas. Type II tumors are high-grade serous ovarian carcinomas (HGSOCs) and represent the majority of EOC cases. Mutational [1] analysis of EOC revealed diverse genetic landscapes that characterize each tumor subtype (Table 1). Endometroid tumors are mainly characterized by phosphatase and tensin homolog (*PTEN)* and catenin beta 1 *(CTNNB1)* mutations [2,3,4,5], whereas mucinous tumors display a high-degree of tumor protein P53 *(TP53)* mutations (52%) [6,7,8] and additional mutations in Kirsten rat sarcoma viral oncogene homolog (*KRAS)*, serine/threonine protein kinase B-Raf *(BRAF)*, and cyclin-dependent kinase inhibitor 2A *(CDKN2A)*. AT-rich interaction domain 1A *(ARID1A)* and phosphatidylinositol-4,5-3-kinase catalytic subunit alpha *(PIK3CA)* mutations characterize a majority of ovarian clear-cell carcinomas [4,5,9,10], while low-grade serous ovarian tumors predominantly show alterations in *KRAS* [5,11,12,13,14,15]. In contrast to type I tumors, HGSOCs show relatively low mutational burden with the exception of ubiquitous *TP53* mutations and additional (10%) mutations in DNA repair genes including breast cancer type susceptibility proteins 1/2 (*BRCA1, BRCA2*) [5,11,12,13,14,15]. These tumors are primarily characterized by the high frequency of gene copy number variation (CNV). Despite genetic variability among EOC, these carcinomas can share a common metastatic location, the peritoneum, suggesting that genetic pathways that characterize each tumor subtype activate common cell biology processes that drive EOC dissemination.

### 1.2. EOC Peritoneal Dissemination

EOC dissemination rarely follows an invasion–metastasis cascade where single cells or collective cell populations break through the basal lamina, penetrate surrounding tissues, and intravasate into the vasculature [26,27]. EOC can form loosely attached outgrowths that extend the apical boundary of the tissue mucosa [28]. Outgrowths can completely detach (release) from the mucosa, transit through the peritoneal fluids, and attach to new sites [29] (Figure 1). This unusual route of dissemination is associated with tumor heterogeneity [30], development of resistant disease [31], and abdominal organ obstruction, which is the leading cause of patient morbidity and mortality [32]. Each step of EOC dissemination reflects a unique molecular mechanism and cellular phenotype. Understanding the molecular and cellular determinants of outgrowth formation, release, and interaction with the microenvironment will provide a fundamental framework that is required for the discovery of new therapies aimed at targeting peritoneal dissemination. In the sections below, we provide a description of known cellular and molecular processes that support distinct steps of EOC dissemination (refer to Table 2).

## 2. Outgrowth Formation

Progression of transformed fallopian tube secretory epithelial cells (FTSEC) toward serous tubal intraepithelial carcinoma (STIC) is considered to be an initiating event in a significant proportion of EOC cases [100]. Some regions of STIC lesions can form multicellular loosely attached outgrowths that extend apically into the peritoneal cavity [28]. Immunohistochemical analysis of STICs suggests the possibility that molecular pathways associated with proliferation, DNA repair, cytoskeletal dynamics, and matrix adhesion contribute to outgrowth formation [39,101,102]. Due to limited availability of transformed FTSEC culture models that would faithfully capture outgrowth dynamics, the role of these pathways in EOC outgrowth formation remains unknown. However, experiments involving a combination of live-cell imaging of transformed epithelial cells, other than FTSEC, provided valuable information on how proliferation and regulation of cytoskeletal dynamics and basement membrane adhesion modulate the initiation of apical outgrowths. In this sub-section, we briefly discuss studies that implicate apical extrusion or apical “budding” in malignant outgrowths and the formation of peritoneal metastases.

Epithelial cell sheets maintain an appropriate cell number and healthy tissue function through the apical extrusion of apoptotic or live cells [103,104,105]. Apical epithelial cell extrusion is associated with the activation of non-muscle myosin II (NMMII)-dependent contractility that, through pulling forces on cell–cell and cell–matrix adhesion, facilitates cell de-adhesion from basal lamina and subsequent translocation of cells to the apical portion of the epithelial sheet [33,34]. Not only do normal cells undergo apical extrusion, but oncogene-transformed cells can also translocate on top of the apical surface of the epithelium, forming outgrowths [106]. Src- or Ras-transformed epithelial cell extrusion requires activity of Rho-associated kinase (ROCK) and NMMII [35,36] and is coupled to gap 2/mitosis (G2/M) transition [37], indicating that different oncogenic factors use common pathways that involve the regulation of cell-cycle- and cytoskeleton-mediated modulation of adhesion that is critical for outgrowth formation. Because STICs are proliferative [107] and G2/M progression is associated with reduced adhesion to the basement membrane [38], we suggest that rapidly dividing FTSECs, through the loss of basement membrane adhesion, undergo apical extrusion and continue to proliferate while remaining attached to apical surfaces of the epithelial layer. STIC lesions deposit laminin γ1 on apical and cell–cell junctional surfaces [39]; thus, it is possible that re-attachment of extruded cells to the apical surface of the epithelium is mediated by laminin-binding integrins, such as integrin β1. In favor of this possibility, there are studies implicating integrin β1 activation and extracellular matrix (ECM) deposition along apical and cell–cell junctional surfaces of detached transformed FTSECs and tumors isolated from ascites [40]. Simultaneous examination of integrin β1 activation and surface deposition of laminin γ1 in STICs would provide further information about whether outgrowths, through ECM adhesion, maintain cell-to-cell attachment.

In contrast to apical extrusion, where single cells extrude and give rise to malignant outgrowths, apical budding involves collective translocation of a group of carcinoma cells that colonize peritoneal tissues through their apical surfaces [42]. Live-cell imaging and immunofluorescence analysis of tumors collected from peritoneal fluids of colorectal cancer (CRC) patients revealed that peritoneal dissemination was supported by tumor clusters that underwent apical budding collectively [43]. Similar to apical extrusion of single cells, collective apical budding required NMMII and ROCK activity, further reinforcing a crucial role for cell contractility and ECM remodeling in mediating early steps of peritoneal dissemination. Loss of a cell polarity regulator partitioning defective homolog D (ParD6), inhibition of tumor growth factor beta receptor 1 (TGFBR1), and attenuation of mothers against decapentaplegic homolog 2 (SMAD2) expression accelerates the generation of CRC cell populations capable of apical budding, suggesting that interference with apical–basal polarity drives peritoneal dissemination. The attachment of CRC budding tumor cell populations to the peritoneal wall or an artificially assembled ECM occurs through the engagement of these tumor cell apical surfaces. Because STIC apical surfaces face the peritoneal cavity, it is possible that, similar to CRC, STIC peritoneal dissemination involves collective budding, and apically localized integrins meditate attachment of STIC to proximally located serosal surfaces. It remains to be established, by using appropriate markers of myosin activation, cell polarity, and integrin activation, whether transformed FTSECs that form peritoneal outgrowths activate myosin contractility and use apically localized integrins to adhere to metastatic sites, such as ovarian mesothelium or omentum. Development of new models that recapitulate malignant outgrowth formation by transformed FTSECs would provide opportunities to examine whether apical extrusion or apical budding initiates EOC dissemination, and whether extrusion mechanisms such as ROCK/NMII activation and remodeling of integrin β1 adhesion are required for the initiation of EOC dissemination.

## 3. Release

Cells detach because they lose adhesion to a basement membrane [108]. Upon release from the basement membrane, detached transformed cells either die [109] or continue to grow [110]. EOC outgrowths are often associated with the nearby presence of histologically and molecularly similar free-floating multicellular tumor clusters [111], indicating that outgrowths undergo detachment. The mechanisms responsible for detachment are not well understood. Early experiments utilizing two-dimensional monolayer cultures of ovarian cancer cell lines provided evidence that constitutively active membrane type 1 matrix metalloproteinase (MT1-MMP), through integrin α3 cleavage, promoted EOC monolayer sheet detachment [44]. Al Habyan et al. [45] demonstrated, using live-cell imaging, that ovarian cancer cells detached from a cellular cluster collectively or as single cells. Interestingly, most of the single cells died upon release, indicating that multicellular clusters that detach from spheroids bypass death associated with the loss of adhesion contact. Both of these studies indicated that mechanisms of adhesion regulation intrinsic to tumor cells are essential for the release of tumor clusters into the peritoneum. Microenvironmental factors could also influence how tumor outgrowths are released from the primary site. For instance, the close proximity of tubal mucosa to ovarian surface epithelium allows FTSECs to form adhesions with the mesothelium and grow directly on the surface of the ovary [46]. Recent evidence [112] suggests that individuals who underwent salpingectomy (removal of Fallopian tubes), for reasons other than ovarian cancer, possessed microscopic tubal tissue on the surface of the ovary, indicating the possibility that fallopian tube mucosa naturally comes in contact with the ovarian surface. Thus, it is conceivable to think that malignant outgrowths adhere to ovarian mesothelium before release. Supporting this possibility, there are ex vivo experiments demonstrating collective adhesion of transformed FTSECs to the surface of the ovary [47]. This mode of dissemination would lead to adhesion formation between the ovary and the fallopian tube. Due to fallopian tube movement, the nature of this attachment would be transient, leading to the rupture of tubo-ovarian tissue and release of tumor clusters into the peritoneum. Experiments in mice indicated that oophorectomy significantly decreased peritoneal dissemination, further reinforcing the idea that interplay between the fallopian tube and the ovary constitutes an important factor in supporting detachment and peritoneal dissemination [113]. A combination of tissue-engineering approaches focused on the development of organotypic co-culture systems that incorporate FTSECs and ovarian mesothelium would be required to study, in detail, the mechanism of tubo-ovarian adhesion and its importance in promoting transformed FTSEC detachment.

## 4. Survival and Growth of Detached Tumors in the Peritoneal Cavity

Thirty percent of newly diagnosed ovarian cancer patients, and nearly all recurrent cases present with the abdominal accumulation of multicellular tumor cell clusters, suspended in peritoneal fluids (ascites) [114]. Abdominal ascites accumulates as a result of increased vascular permeability, poor fluid reabsorption through the mesothelium and the lymphatics, and increased peritoneal oncotic pressure. Some of detached tumor cellular clusters suspended in ascitic fluid maintain viability and colonize abdominal organ surfaces through peritoneal fluid movement [115]. Thus, understanding survival mechanisms of detached clusters could potentially help development of new therapeutic approaches that target free-floating carcinoma cell clusters. Early studies explored the hypothesis that ascitic fluid contains biologically active factors that contribute to the survival of detached clusters [116,117,118]. Examination of non-cellular fractions of peritoneal fluid isolated from ovarian cancer patients revealed the presence of mitogenic factors, extracellular matrix components, and a variety of pro-inflammatory molecules [114]. Below, we direct the reader to several studies that highlighted the role of lysophosphatidic acid (LPA), extracellular matrix (ECM) fragments, interleukin 6 (IL-6), and tumor necrosis factor alpha (TNFα), in the regulation of detached tumor survival. We chose to focus on these molecules for two reasons: (i) their concentration levels are higher in EOC malignant ascites as compared to ascites caused by non-cancer-related diseases; (ii) in addition to supporting detached tumor cell survival and growth, these molecules, though regulation of vascular growth factor (VEGF) expression [62,119,120], impact lymphatic and mesothelium permeability [121] and, thus, regulate ascites accumulation and overall ovarian cancer progression [122]. For a more focused review of soluble factors present in the ascitic microenvironments of ovarian cancer patients, we direct the reader to a recently published review by Muller et al. [123].

### 4.1. Role of Lysophosphatidic Acid

Data from multiple laboratories demonstrated a critical pro-survival role for lysophosphatidic acid (LPA), a major mitogen found in ascites and blood of ovarian cancer patients [124,125]. LPA is a water-soluble phospholipid that binds to the LPA receptor, an endothelial differentiation gene family member 2,4,7 (Edg2,4,7) [126,127]. LPA binding to Edg activates heterotrimeric guanine nucleotide binding regulatory proteins (G-proteins). G-proteins are classified into G_s_, G_i_, G_q_, and G_12/13_ and LPA was shown to couple Edg to G_i_, G_q_, and G_12/13_ [48,128,129,130]. The LPA-mediated engagement of G-proteins triggers activation of an array of pro-survival signaling pathways including mitogen-activated protein kinase (MAPK) [49], phosphoinositide 3 kinase (PI3K) [50], protein kinase C (PKC) [51], and small Rho family GTPases Rho [52], Rac [48], and CDC42 [53]. Experiments demonstrated that, in comparison to normal ovarian epithelium, ovarian cancer cell lines displayed increased production of LPA, implicating an intrinsic, oncogene-driven program that regulates autocrine secretion of LPA [131,132,133,134].

The vast majority of EOC cells that initiate from fallopian tube epithelium carry p53 mutations, and, more recently, Chryplewicz et al. [54] elegantly demonstrated that mutant p53 (m-p53)-dependent transformation of FTSECs led to the increased secretion of LPA. The oncogene’s effect on LPA production was associated with downregulation of an LPA-degrading enzyme, lysophosphatidic acid phosphatase type 6 (APC6), and activation of focal adhesion kinase (FAK) signaling. Survival of ovarian cancer cells suspended in a murine peritoneal cavity depended on lower APC6 expression, suggesting the possibility that autocrine production of LPA, through modulation of ECM adhesion, mediates dissemination of detached carcinoma cells. Activation of LPA receptors induces G_12/13_ Rho/ROCK-mediated remodeling of ECM adhesion and increased phosphorylation of FAK [55,56,57]. Therefore, it is possible that engagement of the LPA receptor in suspended EOC cells further reinforces already existing integrin-dependent signaling that is required to sustain survival of detached cells. In favor of this possibility, there are studies implicating LPA-mediated Rho/ROCK activation in the assembly of integrin adhesion in an ovarian cancer model [58]. The evaluation of LPA effects on ECM adhesion under anchorage deprivation and subsequent measurement of suspended cell survival would require a long-term imaging platform capable of simultaneous acquisition, reconstruction, and quantification of ECM adhesion and cellular viability in multiple Z-planes of three-dimensional (3D) cellular clusters.

### 4.2. Role of Extracellular Matrix

One of the characteristics of ascites-derived tumors is their high degree of cell-to-cell adhesion, cell surface deposition of ECM, and integrin activation [40,59,135]. Ascites contain soluble ECM [136,137], and EOC deposits ECM on cell surfaces [27,39,59,60,61], suggesting that, in the suspended environment of a peritoneal cavity, EOC cells engage integrins to organize ECM adhesion and, thus, suppress cell death associated with anchorage deprivation (anoikis). ECM adhesion is required for growth factor-mediated signaling [138]. Therefore, it is possible that ECM adhesion in suspension is vital for mitogenic and pro-survival effects of growth factors present in ascites. Studies from our laboratory used live-cell imaging approaches to examine the role of integrins and integrin-mediated adhesion in ovarian cancer cluster survival in suspension cultures supplemented with epidermal growth factor (EGF) and insulin [27]. We found that de novo engagement of integrins with a self-deposited matrix on the surface of suspended ovarian cancer cells or transformed FTSECs was required for survival and cell-to-cell attachment. Live-cell imaging of EOC cellular clusters treated with various concentrations of integrin β1 or integrin α5 function-blocking antibodies induced cell death within hours, implying that blocking formation of new ECM adhesion in suspended cells leads to rapid cell death. Interestingly, when we performed the same experiment on tumor cells that had formed tightly adhered clusters, cell death was dramatically decreased. These observations indicate that inhibition of integrins would likely affect cells that dynamically remodel ECM adhesion, for instance, cell populations that undergo apical budding. Because not all ascites-derived EOC cells survive in suspension, it remains to be evaluated whether the degree of ECM deposition by the detached cells correlates with the survival of EOC in ascitic environments.

### 4.3. Role of Soluble Immune-Stimulating Molecules

Profiling of cytokines in ascitic fluids isolated from patients uncovered the presence of pro- inflammatory molecules [63,64], among which IL-6 and TNFα were suggested as critical mediators of tumor cell survival [62,65,66,67,68,69]. The concentrations of these cytokines appeared to be significantly higher in ascites than in serum, indicating that the dissemination of EOC is linked to widespread peritoneal inflammation [139,140].

#### 4.3.1. IL-6 and EOC

Increased IL-6 expression in tumor tissues from EOC patients with disease progression predicts chemotherapy resistance and poor patient survival [65,141,142]. Studies from Coward et al. [143] revealed that inhibition of IL-6 with a function-blocking antibody (siltuximab) significantly decreased intraperitoneal tumor burden in mouse xenograft models. IL-6 inhibition decreased macrophage-mediated inflammation and increased apoptosis, suggesting the possibility that IL-6 controls disease progression through modulation of the microenvironment and/or direct effect on tumor cells. Direct regulation of tumor cell survival by IL-6 involves inactivation of pro-apoptotic molecules and regulation of anti-apoptotic protein expression [144,145,146,147,148]. IL-6 binds to a non-signaling alpha receptor (IL-6R) that dimerizes with a beta receptor (gp130) [149]. This association activates Janus kinases (JAKs) that, through phosphorylation of gp130, couple Ras and phosphatidylinositides [70,71]. IL-6-induced activation of phosphoinositide-dependent kinase-1 (PDK1) promotes protein kinase B (Akt) phosphorylation, and subsequent phosphorylation and suppression of pro-apoptotic proteins Bax (Bcl-2-associated X, apoptosis regulator) and Bad (Bcl-2-associated agonist of cell death) [72,73]. In addition to inactivating Bad, IL-6/JNK-mediated activation of the signal transducer and activator of transcription 3 (STAT3) pathway led to increased expression of anti-apoptotic proteins including Bcl-2 and Bcl-Xl [74]. IL-6-mediated STAT3 signaling is supported by the engagement of matrix receptor integrin β1 [150], suggesting the possibility that maintaining ECM adhesion by detached cellular clusters defines anti-apoptotic programs evoked by IL-6.

#### 4.3.2. TNFα and EOC

Another pro-inflammatory molecule present in peritoneal fluids isolated from EOC patients is TNFα [151]. In spite of its name, “tumor necrosis factor”, which would indicate tumor-suppressive activities, there is mounting evidence that TNFα can promote tumor survival and growth through its pro-inflammatory effects [76,77,78,79,80,81,82,83]. Notably, in EOC models of peritoneal dissemination, where human tumor cells grown in mouse peritoneum constitutively secrete TNFα, genetic attenuation of TNFα expression in tumor cells significantly decreased carcinoma burden [84]. The effect of TNFα produced by cancer cells correlated with increased expression levels of other cytokines including IL-6 and chemotactic factors such as C–C motif chemokine ligand 2 (CCL2) and C–X–C motif chemokine ligand 12 (CXCL12). These results support the hypothesis that TNFα-producing EOC cells regulate cytokine/chemokine networks that directly, or through the microenvironment, support tumor survival and growth. To address the contribution of the microenvironment, Charles et al. [85] examined peritoneal tumor growth under conditions of intact or TNF receptor 1 (TNFR1)-null mutant bone marrow-derived immune microenvironment. The study provided compelling evidence that co-culture of tumor cells and TNFR1-null bone marrow-derived leukocytes, within the peritoneum of a syngeneic murine host, resulted in a significant suppression of carcinoma growth. Interestingly, re-expression of TNFR1 in cluster of differentiation 4-positive (CD4^+^), IL-17-secreting T helper cells, but not macrophages or dendritic cells, rescued carcinoma progression indicating a role for immunologic tolerance in driving EOC progression. Yin et al. [86], however, made a case for tumor-associated macrophages (TAMs) in supporting carcinoma cell proliferation within suspended cellular clusters. The tumor-promoting activities of TAMs depended on direct interaction with EOC cells and local production of EGF. Interestingly, immunofluorescent evaluation of cancer cells and macrophages confined to the same cellular clusters revealed that macrophages occupied central areas of a cellular cluster, and that proliferating carcinoma cells were predominantly present at the cluster periphery. These results suggested that, in addition to proliferation, TAMs promote tumor cell motility through EGF secretion. What would be the advantage of tumor cell motility within a suspended cellular cluster? The effect of TAMs on cancer cell translocation could possibly contribute to tumor cell budding, where cells translocate out of the epithelial mass to initiate new metastatic outgrowths. Previous studies utilized two-photon live-tissue microscopy to capture the effect of EGF-secreting TAMs on tumor cell phenotypes [87,88]. The data suggest that TAMs promote motility of carcinoma that eventually intravasate into the vasculature. In summary, the suspended, pro-inflammatory environment of ascites provides a rich soluble and cellular environment to support EOC survival, growth, and colonization of secondary organs within the peritoneal cavity. Development of imaging approaches to capture the behavior of tumor and stromal cells, in time and space, confined to suspended cellular clusters will broaden our mechanistic understanding of detached tumor biology and help identify molecules that perturb tumor–microenvironment interactions.

## 5. Clearance of the Mesothelium

The hallmark of disseminated EOC is the presence of carcinoma deposits on the serosal (outermost lining) surfaces of the omentum, peritoneal wall, and bowel [94]. Mesothelial cells are the major cellular components of the serosa, and the attachment of tumors to mesothelial cell layers is associated with the absence of mesothelial cells beneath the tumor mass, hinting at carcinoma-mediated displacement or clearance of the mesothelium [152,153,154,155]. Tumor-associated cell surface receptors including CD44 [89,90,91], mucin 16 (MUC16) [92,93], placental cadherin [156], and various integrins [89,155,157,158,159] were all shown to mediate initial adhesion of cancer cells to mesothelial cell layers. More recently, studies indicated a critical role for α5β1 integrin signaling in mediating colonization of the omentum [155,158] and mesothelial clearance [160,161,162,163,164,165,166,167,168]. Live-cell imaging approaches of EOC and mesothelial cells revealed that tumor-mediated mesothelial clearance was regulated by α5β1 integrin-dependent coupling of the tumor cell actomyosin network to the mesothelial cell-derived fibronectin matrix. EOC associated with lateral surfaces of the mesothelial cells and, through physical pulling on fibronectin, induced mesothelial cell movement that resulted in clearance of mesothelial cells from beneath the EOC cells. The carcinoma’s ability to pull on a fibronectin matrix was mediated by NMMII activity, where signaling pathways that activate NMMII are requisite regulators of mesothelial clearance. ROCK, through phosphorylation of non-muscle myosin light chain (MLC), regulates NMMII activity [169], and studies by Kwon et al. demonstrated the vital role of ROCK in ECM remodeling by EOC cells [170]. Interestingly, tumor cells that remodeled ECM in a ROCK-dependent manner also expressed markers of mesenchymal transcriptional programs, suggesting the possibility that mesothelial clearance is regulated by tumor-associated programs that contribute to epithelial-to-mesenchymal transition (EMT). Davidowitz et al. showed that a tumor’s ability to clear mesothelial cell monolayers correlated with expression levels of mesenchymal markers, such as vimentin, and that attenuation of mesenchymal transcription factor expression levels in EOC compromised mesothelial clearance [167]. These studies further supported the role of EMT programs in EOC penetration of the mesothelium. In spite of histologic and electron micrographic data providing evidence of mesothelial cell retraction in response to tumor adhesion and absence of mesothelial cells under the tumor mass [152,153,154,155], a recent report from Pakula and colleagues questioned a tumor’s ability to clear the mesothelium in vivo [171]. Their argument is supported by still images of peritoneal metastasis growing beneath the mesothelial cell layer. These observations may report on a post-invasion event, which is the growth of tumors at a secondary tissue. A plausible hypothesis could be that mesothelial cells, upon tumor cluster invasion, close the cleared area and repair the tissue on top of tumor implants, creating a new serosal layer. Application of in vivo live-tissue microscopy would be needed to capture and quantify tumor–mesothelial cell dynamics and, thus, validate in vitro findings. Nevertheless, growing evidence suggests that EOC cells interact with tissues that are normally covered by a mesothelial layer, indicating that breaching the mesothelial cell layer allows EOC cells to remodel the underlying matrix and initiate new heterotypic interactions with the microenvironment [172]. Cancer-associated fibroblasts, neutrophils, macrophages, and T-cells were shown to play a definite role in providing support for EOC growth at the metastatic sites [123]. Understanding the complexity of EOC and tumor microenvironment interactions will provide new avenues for the development of strategies that restrain tumor growth and survival through targeting tumor microenvironments.

## 6. Metastatic Tumor Microenvironment

It is well established that disseminated EOC adapts to metastatic environments by co-opting non-malignant cells that support tumor attachment, growth, survival, and immune evasion [173]. One of the most frequent EOC metastatic sites is the omentum [94,95], a thin spongy fatty tissue that covers abdominal organs. Studies using EOC patient omental tissue explants, adapted to ex vivo 3D culture conditions, documented the existence of an intimate relationship between omental cell populations and tumors. Mitra et al. provided evidence that the secretory and pro-inflammatory nature of ovarian tumors reprograms omental fibroblasts into cancer-associated fibroblasts (CAFs) that secrete growth and chemotactic molecules to support tumor progression [96]. The reprograming of normal fibroblasts occurred through the cytokine-dependent regulation of microRNAs (micRNAs) that supported fibroblast cell association with tumors. In another study, Neiman et al. used omental explants and genetically engineered mice to demonstrate that adipocytes that reside within the omentum secrete adipokines to attract EOC to the omental surface [99]. Following EOC attachment to the omentum, adipocytes were found to interact with EOC and subsequently activate fatty-acid breakdown mechanisms (lipolysis). Adipocytes, through fatty acid-binding protein 4 (FABP4), transferred processed fats to tumors for β-oxidation to generate energy required for carcinoma growth. The hypothesis that omental adipocytes support tumor growth was also explored by Miranda and colleagues [95]. Their studies elegantly demonstrated the effect of adipocytes on activation of the major serine/threonine protein kinase, salt-inducible kinase 2 (SIK2), in EOC tumors. Activation of SIK2 led to stimulation of phosphoinositide 3 kinase (PI3K) and its target Akt, a major signaling pathway involved in the regulation of cell metabolism, survival, proliferation, and motility.

Fibroblasts and adipocytes reside beneath a single layer of mesothelial cells covering the omentum. In order to facilitate initial adhesion of tumor cells to omentum, EOC cells were shown to release transforming growth factor β (TGFβ) and reprogram mesothelial cells to produce, secrete, and deposit fibronectin that, in turn, facilitated tumor attachment [97]. Secretion of fibronectin by mesothelial cells depended on activation of Rac-GTPase in mesothelial cells. In another study, a similar mode of paracrine effect of tumor cells on omental immune microenvironment was observed. Lee and colleagues showed, using in vivo and in vitro models of omental microenvironment–EOC interactions, that tumors secrete cytokines prior to dissemination to attract neutrophils and promote their death [98]. The death of neutrophils was associated with netosis: a process in which neutrophils expel chromatin content into the extracellular space to create sticky neutrophil extracellular traps (nets) of a DNA and protein matrix that is designed to capture and inactivate pathogens. However, in the case of EOC dissemination, the nets successfully captured tumor cells; however, instead of killing, they reinforced carcinoma adhesion and growth. Since the netosis occurred prior to EOC attachment to the omentum, it remains to be answered whether, in response to EOC-secreted cytokines, neutrophils extrude through the mesothelium and subsequently undergo netosis, or the nets are formed beneath the mesothelial cells and directly impact mesothelial cell integrity. Both scenarios would suggest the hypothesis that neutrophils, either before or after netosis, regulate cell–cell and cell–matrix adhesion of mesothelial cells. Application of live-cell microscopy that would reveal the dynamics of mesothelial cell adhesion in response to neutrophil activation or nets could test this hypothesis.

## 7. Limitations of the Review

In such a brief review of a vast subject, even devoid of clinical content and just from the perspective of cell biology, we regret merely representing an exiguous portion of the many valuable contributions from innumerable researchers in the field. We can only hope to invite a broader dialogue and disseminate interest among the cell biology community to address this devastating disease.

## 8. Conclusions and Future Directions

Significant progress was made in developing models that recapitulate phenotypes associated with distinct steps of EOC peritoneal dissemination. Notably, 3D culture approaches that incorporate multiple cell types or even whole-tissue explants proved to be crucial in studying complex molecular mechanisms of mitogenic signaling, cell–cell and cell–matrix adhesion, metabolism, and immune response. Furthermore, application of live-cell imaging brought a spatial–temporal dimension to the study of phenotypes associated with EOC dissemination. These cell biology assays provide an excellent tool for researchers to discover new molecular mechanisms and carcinoma cell behaviors. In Figure 2, we present a graphical scheme of EOC dissemination from tubal surfaces, highlighting regulators of outgrowth formation, detachment, survival, and colonization. In the era of single-cell analysis of a tumor and its microenvironment, new regulators of distinct steps of EOC metastasis will be identified. Thus, further development of cell biology organotypic assays will become an important tool for the evaluation and validation of new targets.

There is still a gap in knowledge regarding the biology related to EOC-mediated obstruction of organs within the peritoneal cavity. Bowel obstruction is a frequent cause of death among EOC patients [174,175,176], yet the molecular and cellular mechanisms of EOC-mediated bowel obstruction are unknown. The mesothelial cell layer directly covers smooth muscle cells of the small bowel. EOC tumors that colonize small bowel serosa obstruct the function of intestinal smooth muscle cells through unknown mechanisms. Development of new cell biology approaches that include co-culture of mesothelial and intestinal smooth muscle cells or intestinal tissue explants will have to be engineered, in order to investigate obstruction mechanisms. Moreover, the application of advanced imaging technologies capable of quantification of tumor and stromal phenotypes, in real time and 3D space, will be required to monitor carcinoma and its microenvironmental interactions.

Identification of FTSECs as cells of origin for a significant proportion of EOCs propelled research toward understanding the natural history of the disease with hopes for discovering cell phenotypes associated with mechanisms of outgrowth formation and tumor detachment. Using appropriate organotypic models of FTSECs, we will be able to understand mechanisms of tumor initiation and find strategies to detect and eliminate early tumors.

## Figures and Tables

**Figure 1 cancers-11-01957-f001:**
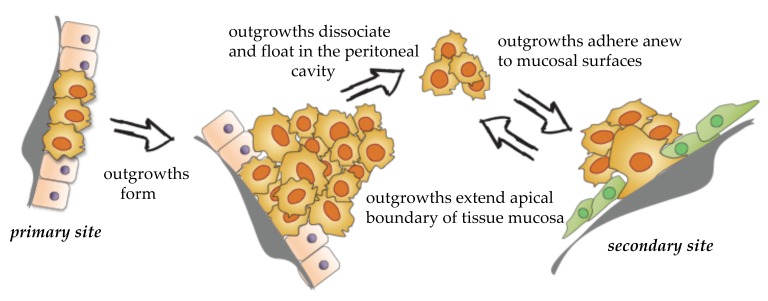
EOC outgrowth formation, dissociation, and colonization.

**Figure 2 cancers-11-01957-f002:**
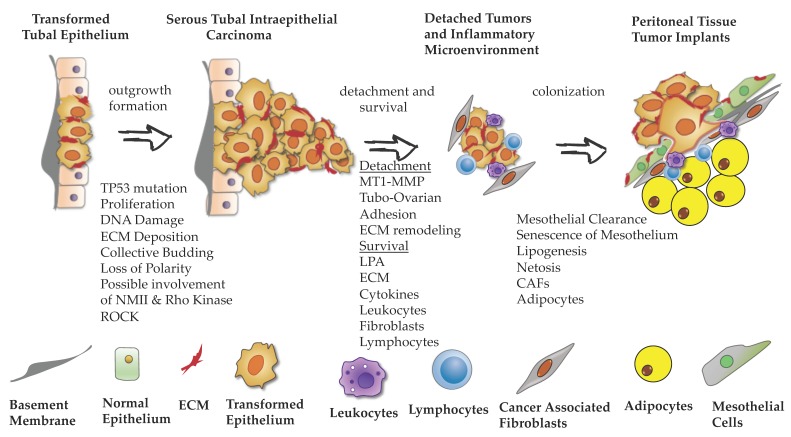
Stages of EOC metastasis.

**Table 1 cancers-11-01957-t001:** EOC subtypes with reported mutations and metastatic sites.

EOC Main Types	EOC Subtypes	Mutations [1]	Metastatic Sites	References (PMID)
**Type I**	Endometroid	*PTEN*, *CTNNB1, PPP2R1α,* MMR-deficient, *ARID1A*	Distant lymph node metastasis, liver parenchymal metastasis, plural effusion with positive cytology	[2,3,4,5]
	Mucinous	*TP53, KRAS*, *HER-2* amplification	Peritoneum, omentum, appendix gastrointestinal, pancreas, cervix, breast, uterus Distant lymph node metastasis, liver parenchymal metastasis, plural effusion with positive cytology	[6,7,8,16,17,18,19]
	Clear cell	*PIK3CA*, *KRAS*, *PTEN*, *ARID1A*	Peritoneal cavity, paraaortic lymph node, distant metastasis in parenchymal organ; Pleura, liver, lung, may initially present with bone metastases, and skin metastases very rarely	[4,5,9,10,20,21,22]
	Low-grade serous	*BRAF*, *KRAS, NRAS*, *ERBB2*	Distant lymph node metastasis, liver parenchymal metastasis, plural effusion with positive cytology, bone	[5,23,24,25]
**Type II**	High-grade serous	*TP53*, *BRCA1*, *BRCA2*, *CDK12*	Distant lymph node metastasis, liver parenchymal metastasis, plural effusion with positive cytology, omentum, falciform ligament, sigmoid serosa, appendix, pelvic side wall, paracolic gutter, bladder serosa	[5,11,12,13,14,15]

**Table 2 cancers-11-01957-t002:** EOC dissemination steps driven by cellular and molecular mechanisms.

Dissemination Steps	Cellular Process	Molecular Process	References
**Outgrowth Formation**	Modulation of adhesion mediated by cytoskeleton and cell-cycle regulators	-NMMII and ROCK	[33,34,35,36,37,38]
		-Cell arrest at G2/M	
	ECM remodeling	-Activation of MMP, integrin B1, and Lamininγ1 deposition on cell surface.	[39,40,41]
	Loss of apical–basal cell polarity	-Loss of ParD6 (cell polarity regulator)	[42,43]
		-Inhibition of TGFBR1, downregulation of SMAD2	
**Release**	Loss of adhesion to basement membrane	-MT1-MMP by cleavage of integrin α3	[44]
	Escaping anoikis	-Detaching as clusters help bypassing anoikis	[45]
	Proximity of tubal mucosa to ovarian surface epithelium favors direct adhesion		[46,47]
**Survival and Growth of Detached Tumors**	LPA -induced survival signaling	-Activates MAPK, PI3K, PKC, Rho-GTPase, RAC, CDC24	[48,49,50,51,52,53]
		-Downregulation of APC6 (LPA-degrading enzyme)	
		-Activation of FAK signaling	[54,55,56,57]
		-Rho–ROCK-mediated ECM remodeling and assembly of Integrin adhesion	[57,58]
	Adhesion to ECM	-ECM deposition on cell surface with help of upregulated integrins and suppressed anoikis	[27,39,40,48,59,60,61]
		-Required for growth factor-mediated signaling	
	Soluble immune-stimulating molecules	-IL6; inactivation of pro-apoptotic factors, i.e., JAK, RAS, PDK1, AKT, and apoptotic factors, i.e., BAX, BAD	[62,63,64,65,66,67,68,69,70,71,72,73,74,75]
		-Expansion of cancer stem cells after chemotherapy	
		-TNFα; promotes tumor survival and growth, correlated with other cytokines (IL6) and chemotactic factors, i.e., CCL2 and CCLX2	[76,77,78,79,80,81,82,83,84,85]
		-EGF; secreted by TAM, promotes cell mobility.	[86,87,88]
**Adhesion and Clearance of the Mesothelium**	Appropriate niche for adhesion of suspended cancer cells through cell surface receptors	-Cell-surface receptors; CD44, MUC16, placental cadherin, integrins such as α5β1	[89,90,91,92,93]
		-Requires activation of NMMII and ROCK	
		-Mediated by EMT; upregulation of vimentin	
**Metastatic Tumor Microenvironment**	Tumor cells reprogram non-malignant cells such as fibroblasts, neutrophils, mesothelial cells, adipocytes by secreting pro-inflammatory molecules	-Fibroblasts reprogramed by cytokine-dependent regulation of miRNAs, turn to CAF and secrete growth and chemotactic molecules to support tumor progression	[94,95,96]
		-Mesothelial cells reprogramed by TGFβ secreted from tumor cells, secrete more fibronectin, facilitate tumor attachment	[97]
		-Tumor cells secrete cytokines to attract neutrophils and promote their death and netosis, creating nets that capture and reinforce adhesion and growth of tumor cells	[98]
		-Adipocytes secrete adipokines to attract cancer cells to the omental surface. Activate lipolysis in cancer cells which provide energy for cancer growth. Adipocytes also activate kinases, including SIK2, leading to PI3K/AKT axis, which regulates cell survival, proliferation, and motility.	[95,99]

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
