# Peer review of "Ovarian Cancer Dissemination—A Cell Biologist’s Perspective"

_cancers, 2019, doi:10.3390/cancers11121957_

Round 1
Reviewer 1 Report
The authors have made an effort to restructure the manuscript and have achieved an improved focus. While the later part of the paper is still rich in detail and not easy to read for lack of "bottom-line" conclusions, it does accurately reflect the mechanisms elucidated in the technical literature.
Reviewer 2 Report
The review is substantially improved, the authors replied to all my comments.
This manuscript is a resubmission of an earlier submission. The following is a list of the peer review reports and author responses from that submission.
Round 1
Reviewer 1 Report
The article “Ovarian Cancer Dissemination – a Cell Biologist’s Perspective” by Farsinejad et al. strives to give a perspective on molecular and cellular processes that contribute to distinct steps of the peritoneal dissemination by ovarian carcinoma.
The article contains 146 references and reflects a substantial exertion to capture the cell biology-relevant literature. Unfortunately, no effort has been made to boil down the available volume of information to common paradigms about ovarian cancer spread. The review reads like a long-winded list of summaries of existing research reports.
In the main text, reference is made to a Figure 1. However, there are no display items or legends in the review copy. Tables or Figures could have helped to boil down the information to some central message on a model for ovarian carcinoma dissemination.
In section 4, the choice to focus on three purported survival and growth mechanisms (LPA, ECM, immune-stimulation) over other candidates needs to be explicitly justified.
As a minor point, 3D cultures are prominently featured in the abstract, although they play a moderate role only in the last paragraph of the treatise.
Overall, there is potential value in the manuscript, but the present writing is insufficiently refined to warrant publication.
Reviewer 2 Report
The review by Farsinejad et all focus on the mechanism through which ovarian cancer cells spread to form metastasis.
The manuscript is interesting, there are few things I would add to have a more comprehensive manuscript:
-a nice image that can describe/summarize what’s written in the text.
-a table, where the authors can include the molecules or pathways involved in the dissemination
- I would probably try to shorten a bit the whole document.